# Combined Treatment with PI3K Inhibitors BYL-719 and CAL-101 Is a Promising Antiproliferative Strategy in Human Rhabdomyosarcoma Cells

**DOI:** 10.3390/molecules27092742

**Published:** 2022-04-24

**Authors:** Manuela Piazzi, Alberto Bavelloni, Vittoria Cenni, Sara Salucci, Anna Bartoletti Stella, Enrica Tomassini, Katia Scotlandi, William L. Blalock, Irene Faenza

**Affiliations:** 1Istituto di Genetica Molecolare ‘‘Luigi Luca Cavalli-Sforza’’, Consiglio Nazionale delle Ricerca (IGM-CNR), 40136 Bologna, Italy; manuela.piazzi@cnr.it (M.P.); vittoria.cenni@cnr.it (V.C.); 2IRCCS, Istituto Ortopedico Rizzoli, 40136 Bologna, Italy; 3Laboratorio di Oncologia Sperimentale, IRCCS, Istituto Ortopedico Rizzoli, 40136 Bologna, Italy; alberto.bavelloni@ior.it (A.B.); katia.scotlandi@ior.it (K.S.); 4Dipartimento di Scienze Biomediche e Neuromotorie (DIBINEM), Università di Bologna, 40138 Bologna, Italy; sara.salucci@unibo.it; 5Dipartimento di Medicina Specialistica, Diagnostica e Sperimentale (DIMES), Università di Bologna, 40138 Bologna, Italy; anna.bartoletti2@unibo.it (A.B.S.); enrica.tomassini2@unibo.it (E.T.)

**Keywords:** soft tissue sarcoma, rhabdomyosarcoma, RTK/RAS/PI3K, PI3K inhibitor, synergism, drug repurposing

## Abstract

Rhabdomyosarcoma (RMS) is a highly malignant and metastatic pediatric cancer arising from skeletal muscle myogenic progenitors. Recent studies have shown an important role for AKT signaling in RMS progression. Aberrant activation of the PI3K/AKT axis is one of the most frequent events occurring in human cancers and serves to disconnect the control of cell growth, survival, and metabolism from exogenous growth stimuli. In the study reported here, a panel of five compounds targeting the catalytic subunits of the four class I PI3K isoforms (p110α, BYL-719 inhibitor; p110β, TGX-221 inhibitor; p110γ, CZC24832; p110δ, CAL-101 inhibitor) and the dual p110α/p110δ, AZD8835 inhibitor, were tested on the RMS cell lines RD, A204, and SJCRH30. Cytotoxicity, cell cycle, apoptosis, and the activation of downstream targets were analyzed. Of the individual inhibitors, BYL-719 demonstrated the most anti-tumorgenic properties. BYL-719 treatment resulted in G1/G0 phase cell cycle arrest and apoptosis. When combined with CAL-101, BYL-719 decreased cell viability and induced apoptosis in a synergistic manner, equaling or surpassing results achieved with AZD8835. In conclusion, our findings indicate that BYL-719, either alone or in combination with the p110δ inhibitor, CAL-101, could represent an efficient treatment for human rhabdomyosarcoma presenting with aberrant upregulation of the PI3K signaling pathway.

## 1. Introduction

Approximately 7% of cancers arising in children and 1% of those arising in adults are soft tissue sarcomas. Of these soft tissue sarcomas, rhabdomyosarcoma (RMS) is the most common, comprising 50% of the soft tissue sarcomas diagnosed in children and adolescents, and characterized by distinctive traits of skeletal muscle lineage [1]. The global disease burden of RMS varies according to geographic location with an incidence of 4.5–5.0 individuals per million in Western cultures (US and Europe) to around 2.0 individuals per million in diverse Asian cultures (Japan, China, India). In addition, odds of developing RMS were found to be significantly lowered if both parents were Hispanic, suggesting potential genetic/ethnic, environmental and cultural/behavorial aspects to disease occurrence [1].

The current histopathological classification comprises five histotypes, but the two most common are referred to as alveolar rhabdomyosarcoma (ARMS) and embryonal rhabdomyosarcoma (ERMS), accounting for about 20–30% and 70–80% of all RMS, respectively [2,3,4]. ERMS is common in early childhood with a second peak of incidence in early adolescence, while the incidence of ARMS is constant from childhood through early adulthood. Pediatric RMS is more common in males than females with a ratio of 1.5:1, while in mid-/late-adulthood, pleomorphic RMS primarily occurs in males [1]. Both genetic and environmental risk factors have been described for disease development, but how much these factors truly contribute to disease is still inconclusive. Although patients with germline syndromes develop RMS more frequently than their normal peers, only about 5% of RMS patients present with a germline syndrome co-morbidity. In addition, studies on environmental factors are very limited, but have suggested associations between RMS and pre-natal X-ray exposure, parental recreational drug use, childhood allergies, and the use of fertility medications, among others [1].

The current molecular classification identifies two major subsets, those harboring either the fused Paired Box 3-Forkhead box O1 (PAX3-FOXO1) or the PAX7-FOXO1 transcription factor generated from recurrent specific translocations (fusion-positive RMS), and those lacking one of these signature fusions (fusion-negative RMS) [4,5,6]. ARMS are typically PAX3-FOXO1 (60%) or PAX7-FOXO1 (20%) fusion-positive RMS characterized by the poorest prognosis; however, promising results to arrest tumor progression have been obtained using targeted inhibitors destabilizing the PAX3-FOXO1 oncoprotein in animal models. In contrast, fusion-negative RMS are characterized by the presence of different alterations, including losses or mutations in the p53 pathway, the RAS pathway, and cell cycle genes, as well as the gain of chromosomes [3,4].

Both fusion-positive and -negative RMS share alterations in genes that provoke the aberrant activation of the receptor tyrosine kinase (RTK)/RAS/phosphatidylinositol-3 kinase (PI3K) axis, such as RAS, phosphatidylinositol-4,5-bisphosphate 3-kinase, catalytic subunit α (PIK3CA), fibroblast growth factor receptor (FGFR)-4, avian erythroblastosis oncogene (ERBB2/HER2) receptor tyrosine kinase and platelet-derived growth factor receptor (PDGFR), as well as insulin-like growth factor (IGF)-2 overexpression [7,8,9,10]. In fusion-positive RMS, PAX3-FOXO1 (or PAX7-FOXO1) down-regulates phosphatase and tensin homolog (PTEN), thus promoting the activity of PI3K, AKT and mammalian target of rapamycin (mTOR) [11]. The PI3K/AKT axis stimulates the uptake of glucose, aminoacids and other nutrients, stimulating metabolic processes that favor a non-differentiated cell state, tumor progression and metastasis [12]. Moreover, the AKT/mTOR pathway, being responsible for the activation of hypoxia-inducible factors (HIFs), promotes tumor survival under low oxygen conditions. The current protocol used to treat RMS involves a multifactorial approach including surgery, radiotherapy, and chemotherapy; however, the survival rates have remained largely unchanged in the past decade [13,14]. Thus, it is imperative that the molecular events that drive tumorigenesis are defined so that more effective, less toxic therapies can be tested.

PI3K/AKT and its associated pathways have been shown to be frequently altered in cancers [15,16]. Preclinical in vitro and in vivo studies have demonstrated that targeted inhibition of the PI3K/AKT pathway enhances cell death in RMS cell lines [10,17]. PI3K enzymes phosphorylate the 3′-OH of the inositol ring of phosphoinositides and are divided into three classes based on structural similarities, common substrate specificity, tissue distribution and mechanism of action [18]. Class I PI3Ks mainly recognize PI(4,5)P2 as a substrate, thus resulting in PI(3,4,5) production, and are mainly activated downstream of RTKs and G-protein coupled receptors (GPCR). Class II PI3Ks have evolved increased specificity towards PI and PI(4)P as a substrate, producing PI(3)P and PI(3,4)P2, respectively; lastly, class III PI3Ks only recognizes PI as a substrate, thus producing PI(3)P. Class I PI3Ks are the best characterized and more often studied, due to their involvement in cancers [17]; they are heterodimers composed of a catalytic subunit of 110 kDa (p100) and a regulatory subunit of 85 kDa (p85). Among the catalytic subunits, four isoforms have been identified, p110α, p110β, p110γ and p110δ. Although human cancers bearing mutations in PI3K catalytic isoform p110α are frequent, those occurring on the catalytic isoforms p110β, p110γ and p110δ are rare; however, these isoforms can be found overexpressed [15,19]. To date, more than 40 PI3K/AKT/mTOR inhibitors have reached the later stages of clinical development, but few, such as Temsirolimus (Torisel^®^, CCI-779; target: mTOR), Everolimus (Afinitor^®^, RAD-001; target: mTOR), Idelalisib (Zydelig^®^, CAL-101; target: PI3Kδ), Copanlisib (Aliqopa^®^, BAY 80–6946; targets: PI3Kα and PI3Kδ), and Alpelisib (Piqray^®^, BYL-719; target: PI3Kα) have been approved for clinical use [15,20,21,22,23,24]. Among these, Idelalisib (CAL-101) was the first PI3K inhibitor licensed for the treatment of cancer [25]. Despite the high level of interest and resources devoted to developing PI3K inhibitors for therapeutic use, there have been numerous challenges. Overall, single-agent PI3K inhibitor regimens have shown modest anticancer effects in preclinical and early clinical phase [26,27]. In addition, many class I PI3K inhibitors have displayed elevated toxicities (liver toxicity, pneumonia, pneumonitis, diarrhea, fatigue, etc.) at clinically effective doses, promoting their discontinuation and withdrawal from phase studies. Thus, combination strategies might represent a way to overcome toxicity problems and limited activity associated with single compounds, opening a new path to advance the use of PI3K inhibitors in clinical treatments [28].

In the study reported here, we analyzed the anti-tumor activity of the individual PI3K class I isoform inhibitors, Alpelisib (BYL-719; p110α), TGX-221 (p110β), CZC24832 (p110γ), and Idelalisib (CAL-101; p110δ), on a panel of RMS cell lines (Figure 1). These compounds were selected primarily for their ability to preferentially inhibit distinct class I PI3Ks. BYL-719 and CAL-101 had the added advantage of receiving approval from the Federal Drug Administration (FDA) and the European Medicines Agency (EMA). Although TGX-221 (PI3Kβ) and CZC24832 (PI3Kγ) have been discontinued in clinical trials, these inhibitors were chosen over similar inhibitors (GSK2636771 and AS-252424, respectively) due to their lower IC50s (5 nM vs. 36–72 nM and 27 nM vs. 30 nM) and a lack of significant cross-reactivity with other class I PI3K isoforms. Cytotoxicity, proliferation, and apoptosis were evaluated to assess the effectiveness of these compounds on tumor growth and progression. In order to determine if combined p110 isoform inhibition could achieve similar anti-tumor associated effects to single p110 isoform inhibition at significantly lower doses of the inhibitors, combination of the two most effective inhibitors, BYL-719 (p110α) and CAL-101 (p110δ) were compared to AZD-8835, a selective bi-specific inhibitor of p110 isoforms α and δ, which is currently in phase I trials. Our results demonstrate that the combined treatment of RMS cells with PI3K p110α and PI3K p110δ inhibitors (BYL-719 + CAL-101) was effective in targeting the PI3K-AKT axis, inducing the arrest of cell proliferation and the activation of caspase-dependent apoptosis, while allowing for the use of reduced concentrations of Alpelisib and Idelalisib.

## 2. Results

### 2.1. Single Treatment of Rhabdomyosarcoma Cell Lines with Alpelisib (BYL-719) and Idelalisib (CAL-101) Reduced Cell Proliferation

Previous studies have shown that treatment of rhabdomyosarcoma cell lines with Buparlisib (BKM-120), a pan inhibitor of class I PI3K enzymes, inhibited cell growth and proliferation [39,40,41], albeit this compound demonstrated high toxicity and poor activity at tolerable doses. In order to evaluate which PI3K isoforms mainly sustained tumor cell proliferation, the action of several isoform-specific PI3K inhibitors on rhabdomyosarcoma cell proliferation, was assessed (Table 1). First, the expression level of class I PI3K isoforms p110α, β, γ, δ was evaluated by immunoblotting analysis on a panel of rhabdomyosarcoma cell lines, RD (embryonal RMS), SJCRH30 (alveolar RMS), and A204 (Figure 2A). Rhabdomyosarcoma cells not only expressed the ubiquitous p110α and β isoforms, but also expressed p110γ and δ isoforms, which are usually only expressed in leukocytes. Although the expression level of class I PI3K isoforms p110β, γ, δ was the same in all RMS cells, p110α was up-regulated in RD cells compared to SJCRH30 and A204. The effect of isoform specific PI3K inhibitors was then tested (Figure 2B). RD, SJCRH30 and A204 were treated with increasing concentrations of Alpelisib (BYL-719), TGX-221, CZC24832 and Idelalisib (CAL-101) for 72 h, then cell viability was evaluated by MTT assay. Alpelisib (BYL-719) treatment significantly affected cell proliferation of all the cell lines tested, confirming that PI3K p110α was one of the isoforms expected to be involved in sustaining RMS cell proliferation. Additionally, Idelalisib (CAL-101) was shown to have an anti-proliferative effect, on RD and A204 cells. Although CZC24832 and TGX-221 were not efficacious in reducing RD and SJCRH30 viability, they were effective on A204 cells. These data demonstrated that in addition to p110α, β, and γ, rhabdomyosarcoma cells expressed the PI3K p110δ isoform, and cell proliferation could be impaired by targeting this PI3K isoform with the p110δ-specific inhibitor, Idelalisib (CAL-101).

### 2.2. The Combined Treatment of Alpelisib (BYL-719) and Idelalisib (CAL-101) Reduce the Viability of RMS Cell Lines In Vitro in a Synergistic Manner

To assess whether PI3K p110α- and δ-isoform combined inhibition could have a synergistic effect on RMS cell viability, RD, SJCRH30, and A204 cells were treated with increasing concentrations of Alpelisib (BYL-719), Idelalisib (CAL-101) or the combination of both inhibitors, for 72 h (Figure 3A). AZD8835, a dual inhibitor of p110α and δ was also used. Cell viability was then analyzed by MTT assay and the IC50 calculated. Cell viability decreased in a concentration-dependent manner in all the RMS cell lines tested. The IC50 values obtained were sufficiently low as to exclude “off-target” effects which become observable under conditions described in previous studies [42]. It was observed that inhibition of p110α alone significantly affected RMS cell viability, with A204 cells being the most sensitive (IC50 = 2.24 μM); the IC50 values for RD and SJCRH30 cells were 7.93 μM and 9.65 μM, respectively. Inhibition of both p110α and δ with AZD8835, had a greater inhibitory effect on cell viability, compared to the respective use of the single inhibitors, BYL-719 or CAL-101, alone (RD IC50 = 7 μM; SJCRH30 IC50 = 6.47 μM; A204 IC50 = 1.28 μM). Remarkably, the combination of Alpelisib (BYL-719) and Idelalisib (CAL-101) provoked an even more dramatic decrease in RD, SJCRH30 and A204 cell viability, with a calculated IC50 of 2.06 μM, 4.91 μM, and 0.5 μM, respectively. A calculation of the effects resulting from the combined treatment of Alpelisib (BYL-719) and Idelalisib (CAL-101) was performed using the software CalcuSyn, to verify the synergic effect observed in MTT assays. To perform this analysis, MTT assays were conducted as previously described [43]. Isobolograms (Figure 3B) demonstrate the effect of the combined treatment. It is assumed that combination index values <1 determine synergistic effects, 1 determines additive effects and >1 corresponds to antagonist effects. As shown, the combination of Alpelisib (BYL-719) and Idelalisib (CAL-101), had a strong synergic effect in all rhabdomyosarcoma cell lines. Therefore, it could be affirmed that in RMS cell lines, the combined treatment of cells with Alpelisib (BYL-719) and Idelalisib (CAL-101) was more effective in decreasing cell viability, compared to both inhibitors alone and to the mixed inhibitor AZD8835.

### 2.3. Effects of the Combined Treatment of Alpelisib (BYL-719) and Idelalisib (CAL-101) on Rhabdomyosarcoma Cell Homeostasis

#### 2.3.1. BYL-719 + CAL-101 Induced Cell Cycle Arrest

Since effects on cell viability, as determined by MTT, may either result from cell death or loss of cellular proliferation, owing to cell cycle arrest or mitochondrial dysfunction, the percentage of viable cells was determined by trypan blue staining in order to confirm the data obtained by MTT assay (Figure 4A). Results obtained after treatment of the three cell lines with the PI3K inhibitors, indicated that all the compounds induced a time-dependent cell death, with a more relevant effect in SJCRH30 cells. Moreover, the effect of the combined treatment of BYL-719 + CAL-101, after 72 h, resulted in a higher percentage of cell mortality. To evaluate whether the increase in cell mortality resulted from the incapacity to overcome cell cycle arrest, an analysis of the cell cycle was performed by flow cytometry, following pharmacological treatment (Figure 4B). Although a single treatment with CAL-101 did not show any effect on the cell cycle, BYL-719 induced an increase in the number of cells in G1/G0 in both RD and SJCRH30 cells (+11% and +10%, respectively). In SJCRH30 cells, combined BYL-719 + CAL-101 treatment further induced cells to arrest in G1/G0 (+20%), even compared to AZD8835 (+16%). No significative changes were observed in A204 cells.

#### 2.3.2. BYL-719 + CAL-101 Sensitize Cells to Caspase-Dependent Apoptosis

Induction of programmed cell death (or apoptosis) has proven to be one of the best methods in treating both hematopoietic and solid cancers as it promotes cell death in a manner least likely to exacerbate the local inflammatory response, a factor often contributing to tumor progression. To determine the efficiency of the class I PI3K inhibitors in promoting apoptosis either individually or in combination, RD, A204, and SJCRH30 cells were treated with BYL-719, CAL-101, BYL-719 + CAL-101 or AZD8835 (Figure 4C). In RD and A204 cells, each of the treatments resulted in an increase in Annexin V positive cells after 24 h; while in SJRH30 cells, treatment with AZD8835 resulted in no significant increase in Annexin V/PI positivity as compared to the untreated control. In each of the tested cell lines, the most apoptosis induction was observed in the samples treated with either BYL-719 alone or in combination with CAL-101. CAL-101 treatment alone demonstrated a modest induction of apoptosis in each of the cell lines, while any appreciable effect of AZD8835 on apoptosis induction was only observed in A204 cells (Figure 4C).

Caspase-3 is a crucial executioner of apoptosis, and once activated it is responsible for cleaving downstream substrates, such as Poly ADP-ribose polymerase (PARP), thereby initiating the apoptotic process. In order to confirm the data obtained by flow cytometry, a caspase-3 activity assay was performed in parallel. Cells were seeded and treated as described above for 48 h, and endogenous caspase-3 activity was measured (Figure 4D). Both RD and SJCRH30 cells showed an increase in caspase-3 activity when treated with Alpelisib (BYL-719), but not with Idelalisib (CAL-101); however, the combined treatment was significantly more effective in promoting caspase-3 activation. These data were further confirmed by immunoblotting RD cell lysates. Only the combined treatment of Alpelisib (BYL-719) and Idelalisib (CAL-101) induced visible cleavage of pro-caspase-3 and -7, indicating the activation of these caspases (Figure 4E). In contrast, cleavage of pro-caspase-3 or -7 was not observed in Western blots of A204 and SJCRH30 protein lysates. The absence of visible cleavage was likely a factor of reduced cleavage in these cell lines as compared to RD cells and a reduced sensitivity of the Western blot assay compared to the flow cytometry-based assays (Figure 4C,D).

### 2.4. The Combined Treatment of Alpelisib (BYL-719) and Idelalisib (CAL-101) Inhibited the Activity of Pi3k Downstream Signaling Effectors

PI3K inhibition affects the activity of several downstream key signaling molecules involved in aspects of physiological and pathological cellular metabolism, such as the protein kinases AKT and glycogen synthase kinase (GSK). To ascertain whether the combined treatment of Alpelisib (BYL-719) and Idelalisib (CAL-101) affected the activity of these kinases, immunoblotting analysis was performed (Figure 5). The inhibition of PI3K p110α with Alpelisib (BYL-719) prevented Akt phosphorylation at Ser473, thus blocking Akt kinase activity, and significantly reduced GSK3α/β phosphorylation on Ser21/9. The inhibition of PI3K p110δ with Idelalisib (CAL-101), on the contrary, did not greatly affect either AKT (Ser473) or GSK3α/β (Ser21/9) phosphorylation. The results obtained with the combined treatment of Alpelisib (BYL-719) and Idelalisib (CAL-101) were comparable with the use of Alpelisib (BYL-719) alone, as well as that of AZD8835, meaning that, in RMS, AKT, and GSK3α/β activity is strongly regulated by PI3K p110α.

## 3. Discussion

Rhabdomyosarcoma (RMS) is a highly malignant and metastatic pediatric tumor that arises from skeletal muscles. Current treatments of RMS involve a multimodal and aggressive approach, combining surgery with high-dose chemotherapy; this strategy has resulted in an increase in the five-year survival rate to over 70% [13]. Nevertheless, negative prognostic factors, such as tumor subtype (alveolar), age (<10 years), and the presence of metastasis at diagnosis, define a subclass of high-risk RMS with a poor prognosis [14]. The PI3K/AKT/mTOR pathway is involved in the onset and progression of several cancers, among which is RMS [11,44,45]. In addition, the PI3K pathway is implicated in resistance to anticancer therapies and targeted agents; therefore, PI3K inhibitors may restore sensitivity to other treatments when administered in combination with other regimens [23,25,26,28,46].

Several PI3K inhibitors have demonstrated promising activity in preclinical models of solid tumors, providing a rationale for their use in the clinic; thus, there are several currently in phase trials, with a handful having already received FDA and EMA approval for the treatment of several cancer types. Although PI3K inhibitors have been shown to prolong progression-free survival (PFS), the therapeutic index is often unfavorable. Clinical efficacy in the single-agent setting has so far been modest with toxicity often observed at clinically relevant doses. Broad inhibition of class I PI3Ks results in an unfavorable safety profile with off-target effects (including mood disorders and liver toxicity) limiting the clinical utility of many previous PI3K inhibitors [47]. Adverse events associated with the clinical use of these inhibitors, such as hyperglycemia, rashes, and diarrhea, are frequently observed in cancer patients undergoing therapy. In particular, hyperglycemia is intrinsically linked to the inhibition of PI3K p110α, a key mediator of insulin signaling. However, recent clinical data have shown that specifically targeting PI3K p110α can improve PFS and clinical benefit. Thus, one potential option to achieve a clinical benefit and reduce the associated toxicities is through a combinational approach.

In this work, the activity of single class I PI3K inhibitors and their use in combination, were tested in RMS cell lines. Our results demonstrated that in addition to PI3K p110α, β and γ isoforms, rhabdomyosarcoma cell lines also expressed PI3K p110δ and that targeting this isoform in conjunction with p110α could represent a novel pharmacological strategy to be pursued. Although treatment with Alpelisib (BYL-719) was able to inhibit the proliferation of all the cell lines tested, indicating that p110α is indeed involved in sustaining tumor cell proliferation, when combined with Idelalisib (CAL-101), a PI3K p110δ specific inhibitor, the Alpelisib–Idelalisib combination showed a stronger anti-proliferative effect with respect to both the use of single inhibitor and the administration of the double inhibitor AZD8835, as demonstrated by the lower IC50. The synergistic effect of BYL-719 + CAL-101 induced cells to arrest in the G1 phase of the cell cycle, but most of all enhanced caspase 3 activation. Furthermore, treatment with Alpelisib (BYL-719) alone or in combination with Idelalisib (CAL-101), determined a reduction in active AKT (Ser473) and a likely enhancement in active GSK3, confirming the use of PI3K p110α inhibitors such as Alpelisib in specifically targeting the PI3K/AKT signaling axis.

Altogether, these results confirmed that combinational pharmacological inhibition of class I PI3K p110α- and δ-isoforms could be an effective therapeutic strategy for the treatment of RMS, reducing the reported side effects while increasing the effectiveness of the treatment. Thus, results from this study open the door to novel pharmacological combinations and dosing schedules that may have fewer off-target effects. Moving this and similar studies into animal models of RMS will be the next step.

## 4. Materials and Methods

### 4.1. Cell Culture and Reagent

The RMS cell lines used in this study, RD (American Type Culture Collection [ATCC^®^] CCL-136^TM^), A204 (A-204; ATCC^®^ HTB-82^TM^), and SJCRH30 (RC13, RMS 13, SJCRH30; ATCC^®^ CRL-2061^TM^), were purchased from the Inter-Cell Line Collection of the National Institute for Cancer Research (IST, Genoa, Italy) and verified as mycoplasma negative. Cell lines were maintained in modified Dulbecco’s medium (DMEM), high glucose, GlutaMAX^TM^ supplement (Gibco, ThermoFisher Scientific, Milan, Italy), supplemented with 10% inactivated fetal calf serum (FCS; Gibco, ThermoFisher Scientific, Milan, Italy). Cells were cultured at 37 °C in a humidified 5% CO_2_ atmosphere.

All PI3K inhibitors listed in Table 1 were purchased from Selleckchem (Munich, Germany) and dissolved following the manufacturer’s instruction, to generate a stock solution (10 mM) which was subsequently diluted with the medium before adding to the cells. Where not otherwise specified, chemicals were purchased from Sigma Aldrich (Milan, Italy), kits and reagents for biochemistry analysis were from Thermo Fisher Scientific (Milan, Italy).

### 4.2. MTT Assay

To test the effects of PI3K inhibitors, RMS cell lines were cultured for 72 h in the presence of the vehicle (DMSO 0.1%) or increasing concentrations (0–20 μM) of each single inhibitor, BYL-719, TGX-221, CZC24832, CAL-101, AZD8835, or in combination (BYL-719 + CAL-101). Cell proliferation was determined using the MTT (3-(4,5-dimethylthythiazol-2-yl)-2,5-diphenyltetrazolium bromide) cell proliferation kit (Roche Diagnostic, Basel, Switzerland), following the manufacturer’s instructions, as previously described [43]. Briefly, 0.5mg/mL of MTT labeling reagent was added to each well and incubated for 4 h. Purple formazan crystals were solubilized by adding 100 μL of the solubilization solution (0.01 M HCl and 10% SDS) overnight. The plate was subsequently read on an Infinite M200 photometer (Tecan Group Ltd., Mannedorf, Switzerland) at a wavelength of 570 nm. Colorimetric readings were normalized against plates of untreated cells under identical culture conditions.

### 4.3. Protein Extraction and Western Blotting

Cells were lysed in a radioimmunoprecipitation assay (RIPA) lysis buffer containing the complete EDTA-free protease and phosphatase inhibitor cocktails and 25 units/mL of the pan-nuclease, benzonase. Cells were lysed by vortexing for 1 h at 10 °C and then cleared of cellular debris by centrifugation at 14,000× *g* rpm for 10 min at 4 °C. The protein concentration of the cleared lysates was determined using the Bradford protein assay reagent (Bio-Rad, Segrate, Italy). Proteins were separated by sodium dodecyl sulfate polyacrylamide gel electrophoresis (SDS-PAGE) on 4–20% gradient gels and immunoblotted. Primary antibodies used were as follows: Cell Signaling Technologies (Beverly, MA): anti-Akt (Cat #4691; 1:1000), anti-phospho-Ser473 Akt (Cat #4060; 1:1000), anti-PI3Kp110α (Cat #4249; 1:1000), anti-PI3Kp110β (Cat #3011; 1:1000), anti-PI3Kp110γ (Cat #5405; 1:1000), anti-phospho-GSK-3α/β (Cat #9331; 1:1000), and anti-GSK-3α/β (Cat #9315; 1:1000); Santa Cruz Biotechnology (SCBT; Dallas, TX): Anti-PI3Kp110δ (Cat. sc-7176; 1:1000) and anti-β-actin (Cat. sc-1616; 1:1000).

### 4.4. Cell Viability

RD, A204, and SJCRH30 cells were plated in six-well tissue culture plates at a density of 2 × 10^5^ cells/well. Cells were treated with 5 μM of either BYL-719, CAL-101, or AZD8835, or the combination BYL-719 + CAL-101 (2.5 μM + 2.5 μM), for 24, 48, and 72 h. Cells in the respective wells were washed with 1X PBS, and resuspended (1.0 × 10^5^ cells/mL) in a buffer solution of 1X PBS, 0.5mM EDTA and 0.2% BSA. Then, 50 μL of the cell suspension was mixed with an equal volume of 0.4% Trypan blue [48]. The solution was mixed thoroughly and allowed to stand for 5 min at room temperature. Then, 10 μL of the solution was pipetted into a disposable chamber slide and inserted into an automated cell counter (Countess^TM^, ThermoFisher Scientific).

### 4.5. Cell Cycle Analysis

To analyze cell cycle distribution, RD, A204 and SJCRH30 cells were treated with 5 μM of either BYL-719, CAL-101 or AZD8835, or the combination BYL-719 + CAL-101 (2.5 μM + 2.5 μM) for 24, 48 and 72 h. Cells were collected, washed once with ice-cold PBS 1X and fixed in ice-cold 70% ethanol at 4 °C for 24 h. The fixed cells were centrifuged at 1500× *g* rpm for 5 min at 4 °C, the cell pellet was washed twice with ice-cold PBS and stained with 0.5 mL FxCycle™ PI/RNase Staining Solution. Cell cycle distribution was evaluated from 10,000 counts using an Attune Nxt Acoustic Focusing Cytometer, equipped with a blue laser (488 nm) (Life Technologies Corporation, Monza, Italy). Data were acquired in list mode using Attune Cytometric 2.6 software.

### 4.6. Apoptosis Assay

RD, A204, and SJCRH30 cells were treated with 5 μM of either BYL-719, CAL-101, or AZD8835, or the combination BYL-719 + CAL-101 (2.5 μM + 2.5 μM) for 24 h. Apoptosis was evaluated by flow cytometry analysis with the Annexin V-FITC/PI Apoptotis Detection Kit (Life Technologies Corporation, Monza, Italy), exploiting the binding of FITC-conjugated Annexin V for the detection of apoptotic and necrotic cells. Secondary staining with Propidium Iodide (PI) allowed for the distinction of early vs. late apoptotic cells. Cell staining was performed according to the manufacturer’s instructions. Fluorescence resulting from FITC and PI was measured at 530 and 620 nm, respectively. Samples were acquired on an Attune Nxt Acoustic Focusing Cytometer (Life Technologies Corporation, Monza, Italy), and data analyzed using Attune Cytometric 2.6 software (Life Technologies Corporation, Monza, Italy).

### 4.7. Fluorimetric Caspase-3 Enzyme Activity Assay

Caspase-3 activity in total cell lysates was determined by a fluorometric EnzChek caspase-3 assay kit (Molecular Probes, Eugene, OR, USA) using 7-amino-4-methylcoumarin-derived substrate Z-DEVDAMC, according to the manufacturer’s instructions, as previously described [43]. Briefly, RD, A204 and SJCRH30 cells were plated at a density of 1.5 × 10^5^ cells/well in a six-well plate and treated with BYL-719, CAL-101 or the combination BYL-719 + CAL-101. After 48 h, the cells were collected, centrifuged, and washed once with 1X PBS. Cells were then lysed with the RIPA buffer. Protein concentration was determined using the Bradford protein assay reagent (Bio-Rad, Segrate, Italy). Then, 50 μg of protein/well from each sample was transferred into a standard black 96-well plate, in triplicate. Samples were mixed with the kit Reaction buffer, and the plate was incubated for 30 min at room temperature, avoiding direct light exposure. Then, 200 μmol/L of Z-DEVD-AMC, a caspase-3 substrate, was added to each well and left to incubate for 1 h at 37 °C in the dark. Assay conditions were standardized in order that the products of the reaction remained in the linear range of detection. The plate was read using an Infinite M200 photometer (Tecan Group Ltd., Mannedorf, Switzerland) at 496/520 nm excitation/emission wavelengths. The sample readings were calculated by subtracting the absorbance of blank samples (background).

### 4.8. Statistical Analysis

Data are presented as the mean ± the standard deviation for the indicated number of independently performed experiments (at least *n* = 2). Analysis of variance (ANOVA) and t-Student’s tests (paired Student’s two-tail t test or unpaired Student’s two-tailed t test) were applied with the purpose of discriminating significant differences among the experimental groups. Statistical significance was determined for *p* ≤ 0.05. All statistical analyses were performed, and all graphs generated, using the Prism v.6.0 software (GraphPad Software, La Jolla, CA, USA). Image J (NIH, Bethesda, MD, USA) was employed to carry-out densitometric analyses.

## Figures and Tables

**Figure 1 molecules-27-02742-f001:**
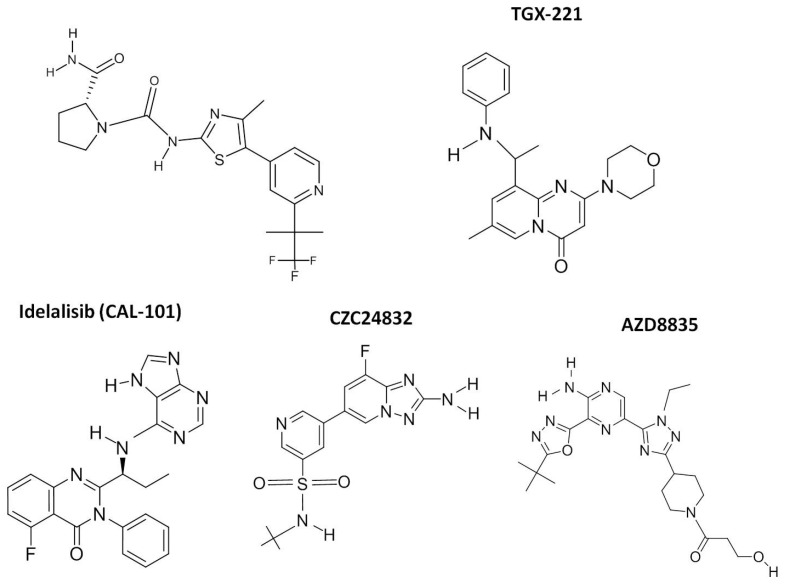
Chemical structure of PI3K class I inhibitors. The chemical structures of the five class I PI3K inhibitors used in this study are presented. Additional details regarding each compound, including the PubChem link are presented in Table 1.

**Figure 2 molecules-27-02742-f002:**
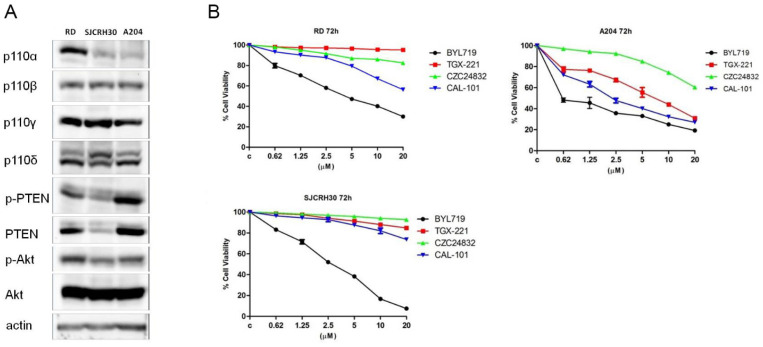
Alpelisib (BYL-719) and Idelalisib (CAL-101) decrease the viability of rhabdomyosarcoma cell lines. (**A**) Western blot analysis of rhabdomyosarcoma cell lines to detect the expression levels of class I PI3K isozymes; 60 μg of protein was blotted in each lane. Antibody to β-actin served as a loading control. (**B**) MTT assay after 72 h treatment with increasing concentrations of BYL-719, TGX-221, CZC24832, and CAL-101. Three replicates per tested concentration and at least three independent experiments were performed (bars, s.d.).

**Figure 3 molecules-27-02742-f003:**
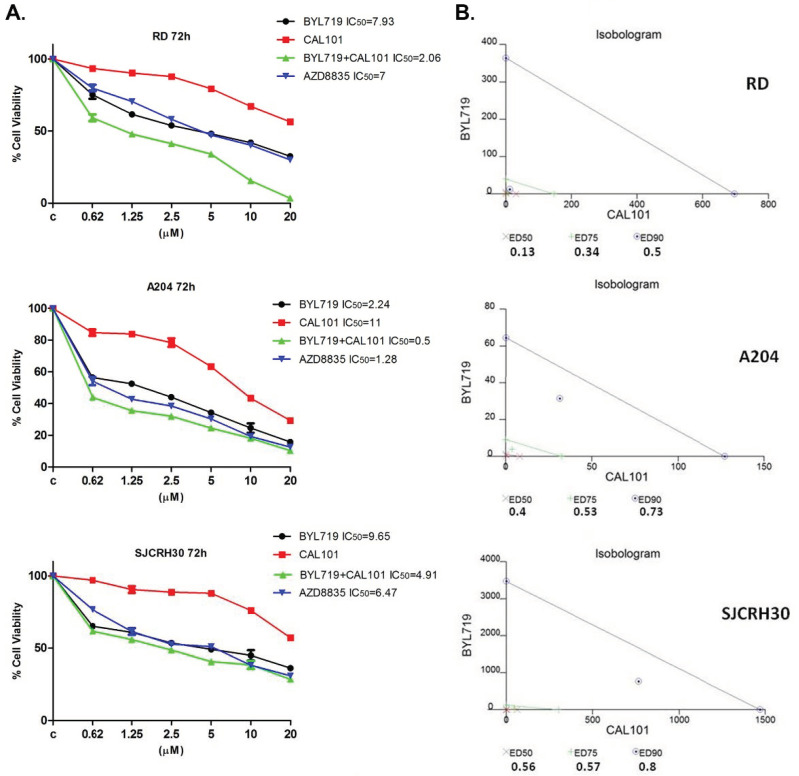
Synergistic effect of BYL-719 + CAL-101 in reducing rhabdomyosarcoma cell viability. (**A**) MTT assay in RD, A204 and SJCRH30 cells treated with BYL-719, CAL-101, BYL-719 + CAL-101, and AZD8835 for 72 h. Cells were seeded in 96-well plates and treated in triplicate with increasing concentrations of inhibitors. After 72 h, a cell proliferation assay was carried-out and the absorbance intensity measured in a microplate reader. Data are representative of three independent experiments. (**B**) Isobologram for combination index (CI) calculations from combined treatment with BYL-719 and CAL-101 for 72 h. The lines link the corresponding concentrations of the two drugs which singularly determine the affected fraction (ED90, ED75, ED50). For each fraction, the corresponding values of CI are reported, with the relative position in the graph (x, +, ⨀).

**Figure 4 molecules-27-02742-f004:**
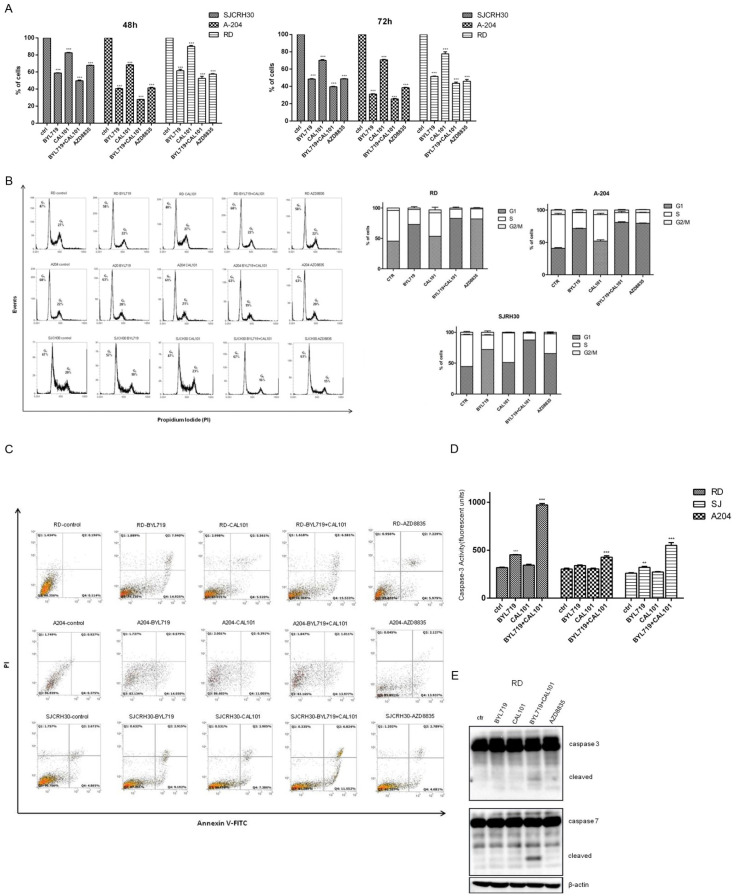
Combination of BYL-719 + CAL-101 arrests rhabdomyosarcoma cell cycle and induces caspase-dependent apoptosis. (**A**) RD, SJCRH30, and A204 cells were seeded in a 6-well plate at a concentration of 2 × 10^5^ cells/well. Cells were treated with 5 μM of either Alpelisib (BYL-719), Idelalisib (CAL-101), AZD8835 or BYL-719 + CAL-101 (2.5 μM + 2.5 μM), or left untreated. After 24, 48, and 72 h, cells were collected and counted. Cell viability was assessed by trypan blue exclusion. Histograms show the percentage of viable cells and are representative of three independent experiments. *** *p* < 0.001. (**B**) Cell cycle analysis of RD, A204 and SJCRH30 cells treated with 5 μM of either BYL-719, CAL-101, AZD8835 or BYL-719 + CAL-101 (2.5 μM + 2.5 μM), or left untreated for 72 h. RD, SJCRH30, and A204 were seeded at the concentration of 1 × 10^6^ cells/well; after 24 h, cells were treated as described above, stained with propidium iodide (PI) and subjected to FACS analysis. Histograms show the percent distribution of cells in each of the cell cycle phases and are representative of three independent experiments. (**C**) RD, SJCRH30, and A204 were seeded at the concentration of 1 × 10^6^ cells in 100 mm plates. After an overnight incubation, cells were treated with 5 μM of either BYL-719, CAL101, AZD8835, or BYL-719 + CAL101 (2.5 μM + 2.5 μM), or left untreated, for 24 h. Cells were then stained with Annexin V-FITC/PI and flow cytometry analysis was carried-out to evaluate the induction of apoptosis. Q1 quadrant (FITC^−^/PI^+^); Q2 quadrant (FITC^+^/PI^+^); Q3 quadrant (FITC^−^/PI^−^); Q4 quadrant (FITC^+^/PI^−^). (**D**) Caspase-3 activity assay in RD, A204 and SJCRH30 cells, treated with 5 μM of either BYL-719, CAL-101 or BYL-719 + CAL-101 (2.5 μM + 2.5 μM), or left untreated for 48 h. Fifty (50) μg of protein lysate for each experimental point was used in the analysis of endogenous caspase-3 activity. Samples were analyzed in triplicate and data are representative of the combined biological experiments. ** *p* < 0.01; *** *p* < 0.001. (**E**) Western blot analysis of RD cells to detect the activation of caspase-3 and caspase-7 after treatment with 5 μM of either BYL-719, CAL-101, BYL-719 + CAL-101 (2.5 μM + 2.5 μM) or AZD8835 for 72 h; 60 μg of protein was blotted in each lane. Antibody to β-actin served as a loading control.

**Figure 5 molecules-27-02742-f005:**
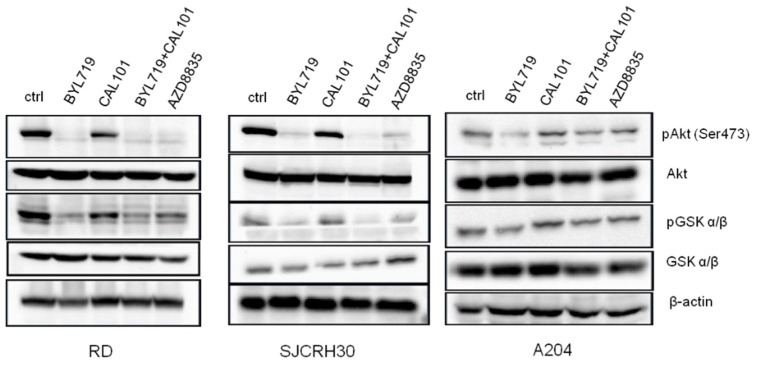
PI3K p110α inhibitors down-modulate Akt signaling. RD, SJCRH30, and A204 cells were seeded at a concentration of 3 × 10^6^ cells in T75 flasks; after an overnight incubation, cells were treated with 5 μM of either BYL-719, CAL101 or BYL-719 + CAL101 (2.5 μM + 2.5 μM), or left untreated, for 24 h. Cells were collected, lysed and protein lysates were subjected to immunoblotting.

**Table 1 molecules-27-02742-t001:** PI3K inhibitors used in this study.

Compound	Alternative Name	PI3K Isoform	Clinical Trials	Cancer	References	Clinical Application
BYL-719	Alpelisib	P110α	Phase 3 Phase 1–2	Breast Cancer Solid Tumors	Pubchem CID: 56649450 [21,29,30,31]	EMA authorized for breast neoplasms 27 July 2020 Commercial name: Piqray
TGX-221	TGX221	P110β	Pre-clinical	Pancreatic Cancer Glioblastoma	Pubchem CID: 9907093 [32]	
CZC24832		P110γ	Pre-clinical	CLL	Pubchem CID: 42623951	
CAL-101	Idelalisib, GS-1101	P110δ	Phase 3	CLL, ALL	Pubchem CID: 11625818 [33,34,35,36]	EMA authorized for lymphomas and leukemias 18 August 2014 Commercial name: Zydelig
AZD-8835	AZD8835	P110α P110δ	Phase1	Solid Tumors and Breast Cancer	Pubchem CID: 76685059 [37,38]	

## Data Availability

The data presented in this study are available upon request.

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
