# Peer review of "Combined Treatment with PI3K Inhibitors BYL-719 and CAL-101 Is a Promising Antiproliferative Strategy in Human Rhabdomyosarcoma Cells"

_molecules, 2022, doi:10.3390/molecules27092742_

Round 1

Reviewer 1 Report

The manuscript described the combined treatment of BYL-719 and CAL-101 as a promising antiproliferative strategy in human rhabdomyosarcoma cells. The manuscript is of high significance. However, I have few suggestions for minor modifications.

  1. The chemical structures of PI3K class I inhibitors (Figure 1) are not clear. The authors are recommended to use some structure drawing tool e.g. ChemDraw.
  2. Rewrite the conclusion of the work.
  3. Write the strong rationale behind selecting only drugs BYL-719, TGX-221, CZC24832 etc.
  4. Rewrite the introduction. Include its epidemiology and global disease burden.
  5. Rewrite the last paragraph of introduction. There should be rationale for combination strategy.

Author Response

We thank the Reviewer for their time and helpful comments.

The manuscript described the combined treatment of BYL-719 and CAL-101 as a promising antiproliferative strategy in human rhabdomyosarcoma cells. The manuscript is of high significance. However, I have few suggestion for minor modifications.

  1. The chemical structures of PI3K class I inhibitors (Figure 1) are not clear. The authors are recommended to use some structure drawing tool e.a. ChemDraw.

Response:

While Figure 1 was produced with ChemDraw using the structures available from Selleckchem (the commercial source of these compounds) and the NCBI's PubChem, at the Reviewer's request, we have looked to improve this Figure by increasing the font of non-carbon atoms and improving its overall appearance.

  1. Rewrite the conclusion of the work.

Response:

Without knowing specifically what the object of the Reviewer's comment refers, we have read through and made edits (adjustments and additions) to the conclusion where we thought best. We hope that we have adequately addressed this point.

  1. Write the strong rationale behind selecting only BYL-719, TGX-221, CZC24832 etc.

Response:

We thank the Reviewer for this suggestion and have add the rationale behind the selection of these particular inhibitors to introduction (lines 147-154 of the revised copy with comments).

  1. Rewrite the introduction. Include its epidemiology and global disease burden.

Response:

As requested by the Reviewer we have added this important information to the introduction (lines 46-69 of the revised copy with comments).

  1. Rewrite the last paragraph of introduction. There should be rationale for combination strategy.

Response:

We thank the Reviewer for suggesting this improvement.  As requested by the Reviewer we have added a statement to the last paragraph of the Introduction to point-out the rational for the use of combinational therapy (lines 156-161 of the revised copy with comments).

Again, we thank the Reviewer.

Reviewer 2 Report

In this study, the authors have looked at the effect of combination treatment of Pi3K inhibitors on growth characteristics of RMS cells. They have done an elaborate study to test the efficacy of multiple inhibitors on RMS growth phenotype. Here are the main concerns that are required to be addressed by the authors:

1) For all the Western blot, authors should submit the original uncut blot and not magnified images of their figures.

2) In figure 3A, A204 cells show more sensitivity to the inhibitor that targets  p110α, where in the blot they show that A204 have lower expression of that isoform as compared to RD. Can the authors explain these results further?

3) In figure 4, the apoptotic FACS assay, the findings suggest early and late apoptosis in RD and SJ cells treated with BYL719 drug alone, but they do not observe any cleaved caspase 3 in their blot for the same group of sample. If cleaved caspase 3 is a standard marker of apoptosis, authors should justify why do they see such discrepancy in their results. Also, they should perform this FACS apoptotic assay including the AZD drug treatment in the cohort.

4) In Figure 4, to strengthen their findings, Authors should include a WB results from the other 2 cell lines for caspase 3  and 7 activation (SJ and A204).

5) The AZD drug seems to be almost as effective as the combination drug treatment in reducing viability, causing cell cycle arrest and also inhibiting Akt activation. Can the authors justify the use of combination therapy vs using AZD drug alone for RMS?

6) In Discussion, line 286, authors mention that broad inhibition of class I PI3K does have side effects and unfavorable safety profile. Their whole study is about inhibiting two isoforms of PI3k, can they predict what kind of toxicity response they might get in their in vivo experiments?

7) In line 277, the statement is not the whole truth. In some of the results, it is suggested that AZD drug does as well as the combination drug treatment. Please modify that statement accordingly.

Author Response

We thank the Reviewer for their time and helpful comments.

In this study, the authors have looked at the effect of combination treatment of PI3K inhibitors on growth characteristics of RMS cells. They have done an elaborate study to test the efficacy of multiple inhibitors on RMS growth phenotype.  Here are the main concerns that are required to be addressed by the authors.

  1. For all the Western blot, authors should submit the original uncut blot and not magnified images of their figures.

Response:

While we agree with the Reviewer that in principle immunoblotting full membranes would be optimal, it has been our standard procedure to cut membranes prior to immunoblotting in order to conserve and make the best use of limited resources.  In order to reduce the possibility of any residual banding arising from immunoblotting with multiple antibodies, all cut membranes were immunoblotted once with the indicated antibody and then discarded (not stripped and re-probed as this tends to give other distortions).  Following incubation with ECL reagent, the images were acquired directly on a BioRad Chemidoc under chemiluminescence acquisition mode.  The images presented are the labeled raw images from blotting, we have no others in our possession. We apologize for this shortcoming.

  1. In figure 3A, A204 cells show more sensitivity to the inhibitor that targets p110α, where in the blot they show that A204 have lower expression of that isoform as compared to RD. Can the authors explain these results further?

Response:

We should first point-out that all the tested cell lines showed more response to the p110α inhibitor than the other tested individual inhibitors (Figure 2B).  This can result from both the properties of the compounds used and the cell's dependence on signaling emanating from p110α.  Of the type I PI3 kinases, activation of p110α is best associated with activation of AKT-mTOR and survival/growth signaling, thus primary sensitivity to p110α inhibition seen in Figure 2B and 3A is not all that surprising.  As visualized in the blot in Figure 2A, A204 cells express the least p110α of the tested cell lines. Due to the importance of p110α, the lower expression would be expected to decrease the IC50 necessary to see an effect if this p110 isoform is truly important for the given cell line; this is what is observed.  In contrast, SJCRH30 cells, which also expresses less p110α versus RD, are more sensitive to p110α inhibition than to inhibition of the other individual p110 isoforms but demonstrate a lower sensitivity than either RD or A204.  This result, though, occurs in a cell background where there is less p-PTEN and p-AKT present (Figure 2A).  The expression level per se is not meant to be indicative of the importance of any given isoform.  The blots in Figure 2A only demonstrate the presence (or absence) of the given p110 isoform.   We apologize for any confusion.

  1. In figure 4, the apoptotic FACS assay, the findings suggest early and late apoptosis in RD and SJ cells treated with BYL719 drug alone, but they do not observed any cleaved caspase 3 in their blot for the same group of sample. If cleaved caspase 3 is a standard marker of apoptosis, authors should justify why do they see such discrepancy in their results. Also, they should perform this FACS apoptotic assay including the AZD drug treatment in the cohort.

Response:

As the Reviewer pointed-out, caspase-3 is considered a standard marker of apoptosis. We believe the issue in this Figure stems from the fact that the assays presented have differing levels of sensitivity. Annexin:PI staining demonstrated appreciable early and late apoptosis in RD and SJ cells after 24 h of treatment with BYL-719 alone or the combination of BYL-719 + CAL-101.  We identified a similar pattern at 48 h using a flow cytometry-based caspase-3 assay.  In both cases, the combination treatment produced the higher level of apoptosis with more dramatic results observed in the RD cell line.  In contrast, western blotting of lysates following 72 h of treatment was only able to identify cleaved caspase -3/-7 in RD cells that were treated with the combination (the sample showing the most cleaved caspase 3 in flow cytometry). These results would suggest that the sensitivity for the western blot assay is significantly below that of the flow cytometry-based assays.

The data presented in Figure 4 indicate that while treatment of the cell lines with the isoform specific inhibitors individually induces apoptosis, combinational treatment with BYL-719 + CAL-101 is more efficient in promoting apoptosis and caspase-3/-7 activation.

We have added the Annexin:PI FACS data for AZD to the figure as requested.

  1. In Figure 4, to strengthen their findings, Authors should include a WB results from the other 2 cell lines for caspase 3 and 7 activation (SJ and A204).

Response:

In the original Figure 4, we only presented data from RD cells at 72 h for the cleaved caspase-3/-7 assays because no cleavage was visible by the western blot assay in the other two lines.  We repeated these experiments with the same results.  We have stated this in the Results (lines 315-322 in the revised copy with comments):

"These data were further confirmed by immunoblotting RD cell lysates. Only the combined treatment of Alpelisib (BYL-719) and Idelalisib (CAL-101) induced  visible cleavage of pro-caspase-3 and -7, indicating the activation, of these caspases (Figure 4E). In contrast, cleavage of pro-caspase-3 or -7 was not observed in western blots of A204 and SJCRH30 protein lysates. The absence of visible cleavage in these cell lines was likely a factor of reduced cleavage in these cells as compared to RD cells and a reduced sensitivity of the western blot assay as compared to the flow cytometry-based assays (Figure 4C and D)."

  1. The AZD drug seems to be almost as effective as the combination drug treatment in reducing viability, causing cell cycle arrest and also inhibiting Akt activation. Can the authors justify the use of combination therapy vs using AZD drug alone for RMS?

Response:

The Reviewer is correct that the profiles between the BYL-719 + CAL-101 and the AZD8835 are similar, but the combination of BYL-719 + CAL-101 was slightly, but consistently, more effective than AZD8835 at a lower IC50.  Thus, the combination of BYL-719 + CAL-101 may achieve similar endpoints as AZD8835 at a lower dose.  Additionally, unlike AZD8835, both BYL-719 and CAL-101 are FDA and EMA approved drugs.

  1. In Discussion, line 286, authors mention that broad inhibition of class I PI3K does have side effects and unfavorable safety profile. Their whole study is about inhibiting two isoforms of PI3k, can they predict what kind of toxicity response they might get in their in vivo experiments?

Response:

This is a very good question and one of the main reasons for undertaking the study.  Class I PI3K inhibitors tend to have the following toxicity responses: hyperglycemia, maculopapular rash, diarrhea, stomatitis and pneumonitis (please see Nunnery and Mayer, Annals of Oncology, (2019)).

Phase trials with BYL-719 or CAL-101 alone, demonstrated that BYL-719 could result in hyperglycemia, diarrhea, kidney toxicity, liver toxicity, rash, pancreatitis and myelosuppression (hyperglycemia was the most prevalent), while CAL-101 was associated with severe diarrhea, liver toxicity, pneumonitis, colon  inflammation, and myelosuppression with increased risk of infection (myelosuppression was the most prevalent).  As the these two compounds have demonstrated prevalently differing toxicity effects, it is our hope (and expectation) that the lower concentrations of each of these compounds, which is necessary to see an affect when used in combination, will have lower to negligible toxic side effects. 

We have included some comment on these aspects (lines 129-136 and lines 358-378).

  1. In line 277, the statement is not the whole truth. In some of the results, it is suggested that AZD drug does as well as the combination drug treatment. Please modify that statement accordingly.

Response:

We believe the Reviewer intended the sentence beginning on 296 and ending on 301.  The Reviewer is correct that in some of the results the AZD drug does as well as the combination of BYL-719 and CAL-101.  The big difference is that the combination showed a response in all tested cell lines at a lower IC50 than that of the AZD drug.  We have modified the statement to reflect this point (line 383-388 in the revised copy with comments).

"Although treatment with Alpelisib (BYL-719) was able to inhibit the proliferation of all the cell lines tested, indicating that p110a is indeed involved in sustaining tumor cell proliferation, when combined with Idelalisib (CAL-101), a PI3K p110δ specific inhibitor, the Alpelisib-Idelalisib combination showed a stronger anti-proliferative effect with respect to both the use of single inhibitor and the administration of the double inhibitor AZD8835, as demonstrated by the lower IC50."

Again, we thank the Reviewer.

Reviewer 3 Report

Hyperactivation of phosphatidylinositol-3 kinase PI3K signaling cascades is a very frequent feature in different human cancers, including rhabdomyosarcoma. It favors the progression and metastasis of the tumor and promotes its survival under hypoxia. Many works have therefore been done on the development of  PI3K inhibitory drugs. Accumulated preclinical and clinical evidence indicate that phosphatidylinositol-3 kinase isoforms  (p110 α, β, γ, δ,) have different functions and are differently activated,  thus, for therapeutic efficacy isoform-selective inhibitors are needed.  

In the submitted paper the therapy of rhabdomyosarcoma has been addressed. In rhabdomyosarcoma cell lines: RD, A204, and SJCRH30 employed as the models, the presence of all four PI3K isoforms was detected by immunoblotting and therefore isoform-selective BYL-719, TGX-221, CZC24832, CAL-101 inhibitors have been evaluated to find out candidate/s clinically efficacious. And for that purpose, cytotoxicity, cell proliferation, and apoptosis were assessed upon treatment with an individual drug or drugs combination. The methods used are sound;  cytotoxicity was checked with MTT assay and trypan blue staining, cell cycle by flow cytometry, caspase-dependent apoptosis by Western blot analyses of the treated cells. The results are presented in 4 Figures. On their basis, the Authors’ conclusion is   BYL-719 (PI3K p110α  inhibitor, Alpelisib), either alone or in combination with CAL-101(PI3K p110δ inhibitor, Idelalisib) could be an effective measure in clinical therapy of human rhabdomyosarcoma. The discussion is concise. Note please, not for all abbreviations, full terms are given

Author Response

We thank the Reviewer for their time and helpful comments.

Hyperactivation of phosphatidylinositol-3 kinase PI3K signaling cascades is a very frequent feature in different human cancers, including rhabdomyosarcoma. It favors the progression and metastasis of the tumor and promotes its survival under hypoxia. Many works have therefore been done on the development of PI3K inhibitory drugs. Accumulated preclinical and clinical evidence indicate that phosphatidylinositol-3 kinase isoforms (p110 α, β, γ, δ,) have different functions and are differently activated, thus, for therapeutic efficacy isoform-selective inhibitors are needed.

In the submitted paper the therapy of rhabdomyosarcoma has been addressed. In rhabdomyosarcoma cell lines: RD, A204, and SJCRH30 employed as the models, the presence of all four PI3K isoforms was detected by immunoblotting and therefore isoform-selective BYL-719, TGX-221, CZC24832, CAL-101 inhibitors have been evaluated to find out candidate/s clinically efficacious. And for that purpose, cytotoxicity, cell proliferation, and apoptosis were assessed upon treatment with an individual drug of drugs combination. The methods used are sound; cytotoxicity was checked with MTT assay and trypan blue staining, cell cycle by flow cytometry, caspase-dependent apoptosis by Western blot analyses of the treated cells. The results are presented in 4 Figures. On their basis, the Authors' conclusion is BYL-719 (PI3K p110α inhibitor, Alpelisib), either alone or in combination with CAL-101 (PI3K p110δ inhibitor, Idelalisib) could be an effective measure in clinical therapy of human rhabdomyosarcoma. The discussion is concise. Note please, not for all abbreviations, full terms are given.

Response:

We thank the Reviewer for bringing this oversight to our attention.  In looking through the manuscript we found that we failed to give the full term for the following abbreviations: ERBB2, GSK, mTOR, PARP, PE, PI, and PTEN.  We have now added the full term to the text.  As the AKT kinase is widely known in the field and is referred to as simply "AKT" in most up-to-date scientific manuscripts, we have not added the full term for this "abbreviation".

Again, we thank the Reviewer.

Round 2

Reviewer 2 Report

I am satisfied with author's responses to my comments. Authors did a great job in providing a detailed explanation to all of my comments and concerns.

Author Response

Reviewer 2:

I am satisfied with author's responses to my comments. Authors did a great job in providing a detailed explanation to all of my comments and concerns.

Response:

We thank the Reviewer for their helpful suggestions and kind comments.
